# Analysis and Control of the Physicochemical Quality of Groundwater in the Chari Baguirmi Region in Chad

**Allaramadji Beyaitan Bantin [1,\*] , Hongping Wang [1,2] and Xia Jun [1]**

1 State Key Laboratory of Water Resources and Hydropower Engineering Science, Wuhan University, Wuhan 430072, China; hongping.wang@whu.edu.cn (H.W.), xiajun666@whu.edu.cn (X.J.)
2 School of Resource and Environmental Sciences, Wuhan University, Wuhan 430074, China
\* Correspondence: bantin2016@gmail.com; Tel.: +86-131-6328-2857

**Abstract:** Water resources are threatened nowadays by pollution that comes from domestic, industrial and agricultural discharges without prior treatment. This pollution causes the degradation of water quality. Surface pollutants can seep through the soil into water tables. The objective of our work is to assess and control the physicochemical quality of the Chari Baguirmi groundwater, to protect human health. The quality of 83 boreholes was assessed, while performing analysis for 12 physicochemical parameters at the National Water Laboratory and at the Center for Quality Control of Foodstuffs in Chad. These parameters are pH, EC, $Ca^{2+}$, $Mg^{2+}$, $Na^+$, $K^+$, $Cl^-$, $SO_4^{2-}$, $NH_4^+$ and $NO_3^-$, $Fe^{2+}$ and $HCO_3^-$. The results obtained were also compared with WHO standards. The geochemical statistical approach has made it possible to characterize hydro geochemical properties and to understand the major processes of mineralization of groundwater resources in the Chari Baguirmi region in Chad. Some of its waters are acidic and weakly mineralized, rich in $Fe^{2+}$ and $NH_4^+$. The origin of the mineralization is due to the alteration of the host rocks and to the hydrolysis of silicate and ferromagnesian minerals, as well as anthropogenic pollution. The ammonium concentrations in N'Djamena boreholes are higher than WHO standards, indicating the presence of pollution which may come from organic waste. These results constitute a preliminary step in understanding hydro geochemical functioning and a basis for monitoring the physicochemical quality of water in the study area.

**Keywords:** analysis; control; physicochemical quality; groundwater; Chari Baguirmi region; Chad

## 1. Introduction

Water is necessary for all forms of life. It is an element for promoting the health of individuals and the socio-economic development of human communities [1].

Without this simple yet complex material, life on earth would never have existed so it is a noble, crucial element that must be protected for future generations. Water intended for human consumption is drinkable when it is free from chemical and biological elements. According to the WHO [2], 1.8 million people a year, 90% of whom are children under five, mostly living in developing countries, die from diarrheal diseases (including cholera); 88% of diarrheal illnesses are caused by poor water quality, poor sanitation and poor hygiene.

The deterioration of the water quality in aquatic ecosystems is an increasingly important issue in Chad as elsewhere in the world and particularly in the Sub-Saharan context where water resources are limited and remain fragile and threatened.

In Chad, water is used primarily in the process of manufacturing food and other products. The agri-food industries of Sarh, Ndjamena and Moundou use different sources of water, but public

water mains and wells are the principle sources of water used by the population. This water can prove to be the main source of contamination and alteration of food.

This work focuses on a study of the physicochemical quality of borehole water in the Chari Baguirmi region, as well as determination of the main chemical facies. It is important to know certain physicochemical parameters which could be the cause of many problems given that poor quality drinking water is a public health problem.

## 2. Materials and Methods

### 2.1. Study Area

Our study area (Figure 1), the Chari-Baguirmi region, is divided into three departments: Baguirmi (city center of Massenya), Chari (city center of Mandelia) and Loug Chari (city center of Bousso), due to decentralization in February 2003. It covers an area of 17,761 km$^2$, with a population estimated at 621,785 inhabitants [3]. This study area covers the localities of the N'Djamena center and its surroundings, Klessoum, Bousso, Ba-illi, Massenya, and Dourbali.

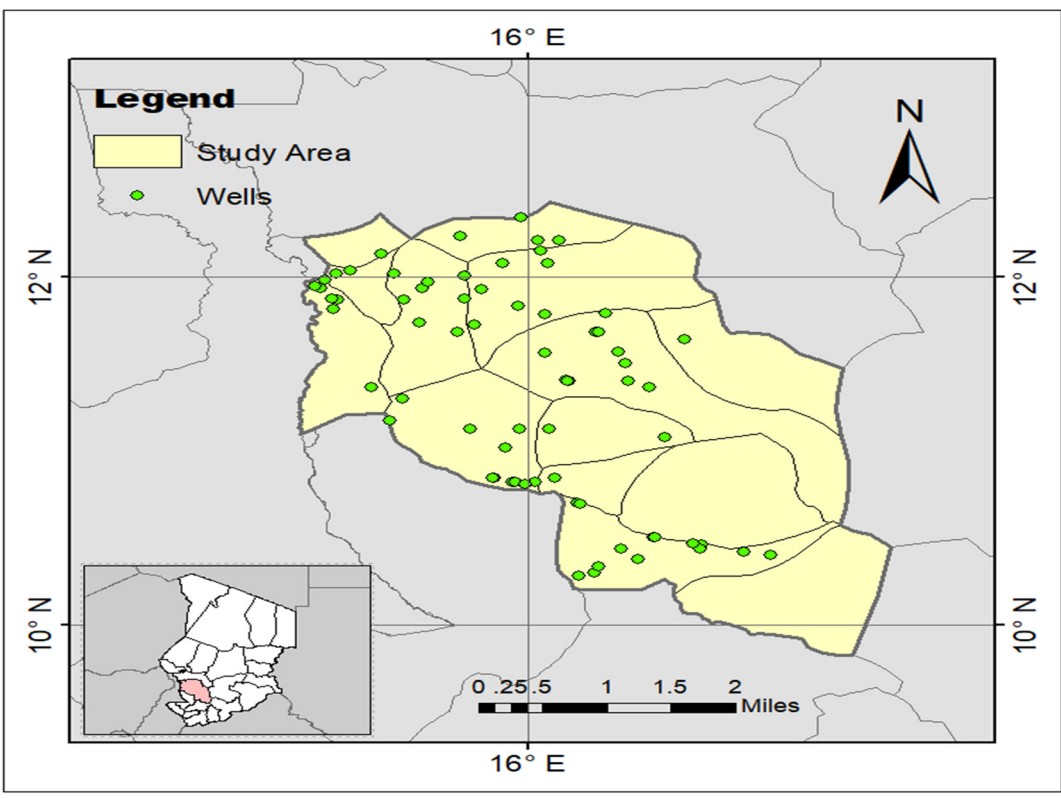

**Figure 1.** Study area.

### 2.2. Analytcal Methods

We determined our physical and chemical parameters in the field (in situ) and in the laboratory according to the technique of Rodier [4].

- **In situ measurements**

The measurements carried out in the field concern physical parameters: temperature (T °C), hydrogen potential (pH) and electrical conductivity (EC). They were measured immediately after taking the water samples, using two portable devices (Xylem Analytics Germany Sales GmbH & Co. KG, WTW, Weilheim, Germany): the WTW pH 330 pH meter (with a precision of 0.01 pH units) and the WTW315i conductivity meter (with a precision of ± 1digit).

- **Laboratory analyzes**

Several analytical methods were used: colorimetric assay methods have made it possible to determine nitrates ($NO_3^-$), sulphates ($SO_4^{2-}$), total iron ($Fe^{2+}$), and potassium ($K^+$), using the DR 2400 spectrophotometer (precision ± 1nm (HACH, Loveland, Colorado, CO, USA)). The volumetric EDTA (Ethylenediaminetetraacetic Acid) method was used to measure calcium ($Ca^{2+}$), magnesium ($Mg^{2+}$) and to determine total duration, using a digital titrator. It was also used to measure chlorides ($Cl^-$), silver nitrate and bicarbonates ($HCO_3^-$) using acid 0.1 N hydrochloric acid from the digital titrator. Sodium ($Na^+$) was determined by flame atomic absorption spectrophotometry and ion selective electrode (ISE) for the measurement of ammonium ($NH_4^+$).

- **Data processing**

The results of the physicochemical analyzes were processed by multivariate statistical analysis methods coupled with hydro chemical methods. The hydro chemical method required the use of a Piper diagram, produced using Diagrams software, for the hydro chemical classification of water. This diagram is widely used in hydrochemistry and gives satisfactory results. The statistical approach is based on the use of Normalized Principal Component Analysis (SNA) and Ascending Hierarchical Classification (CHA) to study the phenomena at the origin of water mineralization, water pooling, and to identify the factors responsible for these groupings. Both of these statistical methods are commonly used in water science. The statistical analyzes were carried out using XLSTAT 2011 software (Addinsoft SARL, New York, NY, USA).

The values of the parameters were compared to the standards of the World Health Organization [5] for drinking water.

## 3. Results

The results of the various physicochemical analyzes carried out on the drilling water of the Chari Baguirmi region are shown in Table 1. This study plays an important role in determining quality, and therefore potability, for use in food and for drinking water. The WHO and European Union drinking water quality standards were used as the basis for the interpretation of our results.

Table 1. Values of the physicochemical parameters of the drilling water analyzed.

| No | Stations | Conductivity | pH | $Ca^{2+}$ | $Mg^{2+}$ | $Na^+$ | $K^+$ | $Fe^{2+}$ | $NH_4^+$ | $NO_3^-$ | $HCO_3^-$ | $Cl^-$ | $SO_4^{2-}$ |
|---|---|---|---|---|---|---|---|---|---|---|---|---|---|
| 1 | TOURLI 1 | 90 | 6.3 | 30.4 | 4.4 | 30.0 | 5.0 | 0.02 | 0.05 | 9.0 | 114.7 | 26.0 | 25.0 |
| 2 | MASSENYA | 102 | 6.4 | 22.4 | 1.9 | 24.0 | 3.0 | 0.06 | 0.15 | 7.0 | 78.1 | 21.0 | 17.0 |
| 3 | MASSENYA | 159 | 6.96 | 20.8 | 1.5 | 21.0 | 2.7 | 0.0 | 0.05 | 8.0 | 70.8 | 19.0 | 15.0 |
| 4 | MASSENYA | 417 | 7.42 | 20.0 | 1.0 | 19.0 | 2.4 | 0.1 | 0.05 | 7.0 | 65.9 | 17.0 | 13.0 |
| 5 | NDJAMENA | 126 | 6.11 | 6.4 | 1.9 | 10.0 | 1.6 | 0.2 | 0.13 | 5.0 | 29.3 | 10.1 | 5.0 |
| 6 | NDJAMENA | 219 | 6.4 | 28.0 | 3.4 | 27.0 | 4.0 | 0.1 | 0.24 | 7.0 | 102.5 | 24.0 | 22.0 |
| 7 | NDJAMENA | 22 | 6.76 | 24.0 | 3.9 | 25.0 | 3.7 | 0.1 | 0.02 | 9.0 | 92.7 | 20.0 | 19.0 |
| 8 | NDJAMENA | 261 | 7.06 | 7.2 | 2.4 | 12.0 | 1.8 | 0.0 | 0.03 | 8.0 | 34.2 | 11.0 | 6.0 |
| 9 | NDJAMENA | 286 | 6.97 | 5.6 | 1.5 | 8.0 | 1.2 | 0.3 | 0.40 | 2.0 | 24.4 | 8.0 | 4.0 |
| 10 | KLESSOUM | 212 | 7.53 | 17.6 | 1.0 | 17.0 | 2.1 | 0.1 | 0.00 | 8.0 | 58.6 | 14.0 | 11.0 |
| 11 | KARNAK | 198 | 6.87 | 11.47 | 2.82 | 3.19 | 3.41 | 0.0 | 3.00 | 0.7 | 173.91 | 0.94 | 0.34 |
| 12 | MASSENYA Q LA PAIX | 181 | 6.53 | 38.63 | 10.17 | 15.18 | 4.73 | 0.0 | 1.81 | 21.8 | 120.8 | 54.51 | 10.49 |
| 13 | TOURLI | 96 | 6.48 | 9.38 | 2.88 | 2.8 | 3.41 | 0.0 | 0.32 | 0.8 | 56.13 | 1.25 | 0.51 |
| 14 | ABOUSSAKINE | 323 | 6.5 | 24.73 | 6.81 | 13.73 | 5.19 | 0.0 | 1.63 | 5.3 | 156.82 | 4.6 | 6.38 |
| 15 | BALABOUDA | 278 | 6.61 | 22.43 | 6.48 | 13.69 | 5.28 | 0.0 | 1.60 | 2.8 | 144.04 | 2.8 | 5.55 |
| 16 | NDJAMENA | 198 | 6.6 | 22.55 | 6.12 | 8.36 | 4.53 | 0.0 | 1.05 | 7.2 | 173.9 | 8.13 | 4.62 |
| 17 | NDJAMENA | 342 | 6.5 | 23.83 | 6.77 | 14.28 | 5.32 | 0.0 | 1.70 | 3.9 | 143.4 | 4.42 | 0.26 |
| 18 | NDJAMENA | 333 | 6.47 | 22.15 | 6.51 | 13.58 | 5.28 | 0.0 | 1.61 | 2.1 | 140.95 | 2.91 | 5.52 |
| 19 | NDJAMENA | 515 | 6.45 | 8.18 | 2.38 | 2.8 | 2.71 | 0.0 | 0.83 | 0.0 | 53.7 | 0.65 | 0.35 |
| 20 | NDJAMENA | 375 | 7.08 | 8.18 | 2.38 | 2.8 | 2.71 | 0.0 | 0.83 | 0.0 | 53.7 | 0.65 | 0.35 |
| 21 | NDJAMENA | 180 | 6.65 | 24 | 13.6 | 3 | 0.8 | 0.0 | 1.31 | 0.0 | 74 | 5 | 4 |
| 22 | NDJAMENA | 366 | 7.2 | 25.6 | 1.2 | 8.6 | 2.4 | 0.0 | 1.01 | 0.0 | 175 | 5 | 1 |

**Table 1.** *Cont.*

| No | Stations | Conductivity | pH | $Ca^{2+}$ | $Mg^{2+}$ | $Na^+$ | $K^+$ | $Fe^{2+}$ | $NH_4^+$ | $NO_3^-$ | $HCO_3^-$ | $Cl^-$ | $SO_4^{2-}$ |
|----|----------|-------------|-----|------|-------|------|-----|------|-------|-------|--------|------|--------|
| 23 | NDJAMENA | 461 | 6.55 | 4.8 | 18 | 2 | 0.4 | 0.0 | 0.24 | 0.0 | 59 | 2 | 0 |
| 24 | NDJAMENA | 590 | 6.6 | 32 | 14.6 | 12 | 3.7 | 0.0 | 0.54 | 0.0 | 233 | 10 | 2 |
| 25 | NDJAMENA | 435 | 6.25 | 40 | 14.6 | 28.3 | 2.4 | 0.0 | 0.77 | 0.0 | 255 | 19 | 30 |
| 26 | NDJAMENA | 758 | 6.92 | 26.4 | 15.6 | 5.7 | 2.8 | 0.0 | 0.26 | 0.0 | 239 | 10 | 15 |
| 27 | NDJAMENA | 231 | 7.13 | 40 | 19.4 | 17.01 | 3.6 | 0.0 | 0.74 | 8.9 | 248 | 20 | 7 |
| 28 | NDJAMENA | 92 | 6.9 | 44.8 | 21.4 | 16.8 | 3.7 | 0.2 | 0.75 | 0.0 | 392 | 17 | 11 |
| 29 | NDJAMENA | 195 | 7.04 | 42.4 | 19 | 17.3 | 3.7 | 0.0 | 0.66 | 0.0 | 137 | 10 | 7 |
| 30 | NDJAMENA | 208 | 6.99 | 45.6 | 35.5 | 27.3 | 4.5 | 0.0 | 1.13 | 0.0 | 143 | 19 | 18 |
| 31 | NDJAMENA | 402.66 | 6.9 | 36 | 7.3 | 6.3 | 2.8 | 0.1 | 1.07 | 0 | 127 | 14 | 2 |
| 32 | NDJAMENA | 146.1 | 7.36 | 47.2 | 10.2 | 23.8 | 3.8 | 0.1 | 0.8 | 1.2 | 128 | 12 | 3 |
| 33 | NDJAMENA | 384 | 6.7 | 36 | 9.2 | 3.7 | 1.8 | 0.5 | 5.14 | 1.2 | 300 | 11 | 6 |
| 34 | NDJAMENA | 327.33 | 7.2 | 32 | 14.6 | 31.6 | 3.6 | 0.6 | 0.77 | 21.7 | 192 | 32 | 12 |
| 35 | NDJAMENA | 146.1 | 7.36 | 20 | 7.3 | 28.4 | 2.6 | 0.1 | 0.48 | 0 | 159 | 10 | 2 |
| 36 | NDJAMENA | 384 | 6.7 | 49.6 | 20.4 | 25.1 | 5.1 | 0.2 | 1.17 | 0 | 162 | 10 | 7 |
| 37 | NDJAMENA | 687.6 | 6.5 | 90.4 | 25.3 | 9.1 | 6 | 0.1 | 0.7 | 0 | 209 | 7 | 1 |
| 38 | NDJAMENA | 270.66 | 6.7 | 60 | 1.2 | 9.5 | 4.3 | 0.1 | 0.88 | 0 | 310 | 10 | 10 |
| 39 | NDJAMENA | 254 | 7.5 | 58.4 | 20.9 | 10.7 | 4.9 | 0.1 | 1.23 | 0 | 127 | 14 | 0 |
| 40 | NDJAMENA | 120.33 | 7.5 | 68 | 30.6 | 8.6 | 2 | 0.2 | 0.01 | 24.9 | 299 | 30 | 13 |
| 41 | NDJAMENA | 461 | 7.6 | 64 | 29.2 | 44.9 | 7.5 | 0.1 | 0.55 | 4.4 | 143 | 40 | 42 |
| 42 | NDJAMENA | 644.33 | 7.4 | 44 | 2.4 | 22 | 2.3 | 0.1 | 0.73 | 5.5 | 133 | 15 | 0 |
| 43 | BOUSSO | 309 | 7.3 | 16 | 5.3 | 13.5 | 3.2 | 0.0 | 0.41 | 6.0 | 87.8 | 7.0 | 5.0 |
| 44 | BOUSSO | 77.15 | 7.5 | 2.4 | 1.0 | 7.1 | 1.0 | 0.0 | 0.33 | 4.0 | 24.4 | 2.0 | 0.0 |
| 45 | BOUSSO | 210 | 7.3 | 88 | 19.4 | 89.0 | 6.0 | 0.0 | 0.71 | 8.0 | 414.8 | 67.0 | 58.0 |

Table 1. *Cont.*

| No | Stations | Conductivity | pH | $Ca^{2+}$ | $Mg^{2+}$ | $Na^+$ | $K^+$ | $Fe^{2+}$ | $NH_4^+$ | $NO_3^-$ | $HCO_3^-$ | $Cl^-$ | $SO_4^{2-}$ |
|----|----------|-------------|-----|-----------|-----------|--------|-------|-----------|----------|----------|-----------|--------|-------------|
| 46 | BOUSSO | 197.7 | 7.3 | 3.2 | 1.9 | 8.0 | 1.5 | 0.0 | 0.29 | 4.4 | 31.7 | 4.0 | 0.0 |
| 47 | BOUSSO | 305 | 7 | 24.8 | 4.4 | 18.0 | 4.0 | 0.0 | 0.19 | 5.6 | 102.5 | 20.0 | 7.0 |
| 48 | BOUSSO | 209.5 | 7.3 | 3.2 | 1.5 | 7.6 | 1.3 | 0.0 | 0.41 | 3.0 | 29.3 | 3.0 | 0.0 |
| 49 | BOUSSO | 199.85 | 7.3 | 15.2 | 5.3 | 18.0 | 3.7 | 0.2 | 0.82 | 3.0 | 97.6 | 10.0 | 6.0 |
| 50 | BOUSSO | 68.9 | 7.6 | 3.2 | 1.5 | 4.7 | 0.5 | 0.2 | 0.66 | 2.0 | 17.1 | 5.0 | 0.0 |
| 51 | BOUSSO | 68.9 | 7.6 | 6.4 | 0.5 | 5.0 | 0.8 | 0.1 | 0.38 | 4.0 | 22.0 | 6.0 | 0.0 |
| 52 | MASSENYA | 247 | 6.57 | 40.35 | 7.9 | 9.08 | 7.65 | 0.0 | 2.74 | 17.1 | 175.73 | 14.78 | 1.94 |
| 53 | MASSENYA | 209 | 5.69 | 80.57 | 17.0 | 7.7 | 7.88 | 0.0 | 0.69 | 3.6 | 128.14 | 34.32 | 2.73 |
| 54 | MASSENYA | 198 | 6.27 | 28.09 | 5.8 | 7.48 | 3.51 | 0.0 | 1.20 | 13.6 | 433.24 | 7.38 | 1.62 |
| 55 | MASSENYA | 305 | 6.26 | 106.52 | 23.8 | 15.2 | 7.36 | 0.0 | 0.00 | 0.3 | 290.45 | 54.45 | 9.54 |
| 56 | MASSENYA | 205 | 5.93 | 37.12 | 8.8 | 14.49 | 4.46 | 0.0 | 2.02 | 0.0 | 162.92 | 22.64 | 6.33 |
| 57 | MASSENYA | 106 | 6.4 | 64.41 | 8.81 | 10.08 | 4.08 | 0.0 | 5.73 | 0.0 | 284.96 | 25.14 | 2.73 |
| 58 | MASSENYA | 969 | 6.53 | 31.3 | 14.8 | 19.75 | 5.66 | 0.0 | 1.11 | 0.3 | 192.82 | 19.56 | 13.62 |
| 59 | MASSENYA | 416 | 6.7 | 30.3 | 4.9 | 18.19 | 5.9 | 0.0 | 2.70 | 0.0 | 98.85 | 1.59 | 0.69 |
| 60 | MASSENYA | 315 | 7.23 | 12.2 | 7.1 | 6.16 | 4.32 | 0.0 | 0.85 | 24.4 | 175.13 | 1.59 | 0.69 |
| 61 | MASSENYA | 196 | 6.76 | 43.7 | 2.6 | 20.21 | 4.43 | 0.0 | 0.64 | 4.4 | 241.03 | 62.97 | 15.4 |
| 62 | BA ILLI | 110 | 5.66 | 1.6 | 1.0 | 2.0 | 0.8 | 0.2 | 0.47 | 5.6 | 7.3 | 2.0 | 8.6 |
| 63 | BA ILLI | 260 | 6.32 | 20.8 | 2.9 | 15.0 | 1.7 | 0.1 | 0.39 | 3.0 | 85.4 | 10.0 | 7.0 |
| 64 | BA ILLI | 143 | 5.26 | 4.8 | 1.0 | 6.3 | 0.7 | 1.3 | 0.39 | 6.0 | 19.5 | 7.0 | 1.0 |
| 65 | BA ILLI | 95 | 6.12 | 6.4 | 1.0 | 7.0 | 0.5 | 0.2 | 0.47 | 5.6 | 24.4 | 6.0 | 0.0 |
| 66 | BA ILLI | 210 | 6.32 | 20.8 | 2.9 | 15.0 | 1.7 | 0.0 | 0.64 | 8.0 | 85.4 | 10.0 | 7.0 |
| 67 | BA ILLI | 481 | 4.96 | 10.4 | 3.4 | 14.0 | 3.2 | 0.0 | 0.55 | 2.0 | 48.8 | 13.0 | 6.0 |
| 68 | BA ILLI | 478 | 4.81 | 4 | 2.1 | 4.0 | 0.4 | 0.1 | 0.39 | 3.0 | 19.0 | 6.6 | 0.0 |

**Table 1.** *Cont.*

| No | Stations | Conductivity | pH | $Ca^{2+}$ | $Mg^{2+}$ | $Na^+$ | $K^+$ | $Fe^{2+}$ | $NH_4^+$ | $NO_3^-$ | $HCO_3^-$ | $Cl^-$ | $SO_4^{2-}$ |
|----|----------|--------------|-----|-----------|-----------|--------|-------|-----------|----------|----------|-----------|--------|-------------|
| 69 | BA ILLI | 51.3 | 5.4 | 1.8 | 0.9 | 2.5 | 0.1 | 0.0 | 0.76 | 8.0 | 7.3 | 3.0 | 0.0 |
| 70 | BA ILLI | 196 | 6.53 | 10.4 | 3.4 | 1.7 | 0.9 | 0.0 | 0.66 | 6.0 | 34.2 | 7.0 | 0.0 |
| 71 | BA ILLI | 537 | 6.52 | 3.2 | 1.5 | 1.9 | 0.7 | 0.0 | 0.82 | 3.0 | 12.2 | 3.0 | 0.0 |
| 72 | BA-ILLI | 341 | 7 | 1.6 | 1.0 | 1.7 | 0.6 | 0.2 | 0.23 | 6.0 | 7.3 | 2.0 | 0.0 |
| 73 | BA-ILLI | 355 | 6.77 | 41.6 | 8.7 | 10.7 | 2.2 | 1.4 | 0.35 | 4.0 | 170.8 | 12.0 | 4.0 |
| 74 | BA-ILLI | 395 | 6.65 | 8.8 | 4.4 | 4.6 | 1.6 | 0.2 | 0.14 | 9.0 | 46.4 | 6.0 | 1.0 |
| 75 | DOURBALI | 498 | 6.8 | 43.7 | 0.47 | 40.25 | 3.53 | 0.0 | 1.25 | 11.0 | 148.8 | 5.88 | 6.62 |
| 76 | DOURBALI | 495.66 | 6.9 | 19.4 | 11.94 | 32.39 | 6.94 | 0.0 | 1.11 | 15.4 | 194.4 | 11.47 | 6.43 |
| 77 | DOURBALI | 499.33 | 6.8 | 12.1 | 4.34 | 24.15 | 5.48 | 0.3 | 0.71 | 4.4 | 168 | 5.29 | 8.72 |
| 78 | DOURBALI | 382.66 | 6.7 | 22.6 | 2.81 | 40.98 | 5.1 | 0.0 | 1.00 | 17.6 | 185 | 29.62 | 6.28 |
| 79 | DOURBALI | 198.5 | 6.8 | 11.1 | 3.32 | 6.12 | 5.1 | 0.0 | 0.88 | 0.0 | 127 | 10 | 14.77 |
| 80 | DOURBALI | 293 | 6.8 | 10.9 | 6.93 | 19.1 | 6.62 | 0.0 | 1.01 | 8.8 | 130 | 12.43 | 3.28 |
| 81 | DOURBALI | 216 | 6.7 | 17.4 | 4.68 | 11.21 | 4.68 | 0.0 | 0.10 | 2.2 | 169 | 4.71 | 0.12 |
| 82 | DOURBALI | 213.9 | 7 | 22.8 | 2.7 | 15.09 | 2.93 | 0.0 | 0.20 | 1.5 | 189 | 1.1 | 14.64 |
| 83 | DOURBALI | 265 | 6.9 | 34.0 | 6.5 | 23.86 | 5.6 | 0.0 | 1.11 | 13.5 | 144 | 14.82 | 1.48 |

The results of the physicochemical analysis presented in this work can be considered admissible, and these results are in agreement with those obtained by Alhabo, [6] and Seid et al. [7].

- **Temperature**

The temperature of water is an important factor as it governs almost all physical, chemical and biological reactions. Any sudden variation in this parameter causes a disturbance in the balance of the water ecosystem and mainly influences climatic variations.

During our sampling, the temperature values were high, but we note that they also varied little from one point to another. The minimum value is 25.8 °C and the maximum 33.1 °C.

- **pH**

pH (Figure 2a) represents the concentration of hydrogen ions in a solution and for natural waters the pH is linked to the nature of the terrain crossed. Certain pH values recorded do not exceed the maximum acceptable pH of the WHO, which ranges from 6.5 to 8.5 for quality drinking water. These waters show notable variations and have a tendency to be acidic, with a minimum pH value of 4.81.

- **Conductivity**

Electrical conductivity refers to the ability of water to conduct an electric current and is determined by the content of dissolved substances, ionic charge, ionization capacity, mobility and temperature. It provides information on the degree of mineralization of the water, varies according to the concentration of dissolved salts and is often influenced by temperature which acts on the dissolution of salts in water [8].

Measuring electrical conductivity makes it possible to quickly, but very roughly, assess the overall mineralization of water and to follow its evolution. The electrical conductivity values obtained vary between 22 μS/cm and 969 μS/cm (Figure 2b). Of the 83 sampling sessions, it should be noted that only the Massenya water point of all those analyzed was weakly mineralized.

The Massenya sample has a higher conductivity value than the other samples, but it is not higher than the WHO standard which is ≤2500 μS/cm.

In this study, the variations observed depend on the aquifer that contains these waters and the temperature of the borehole.

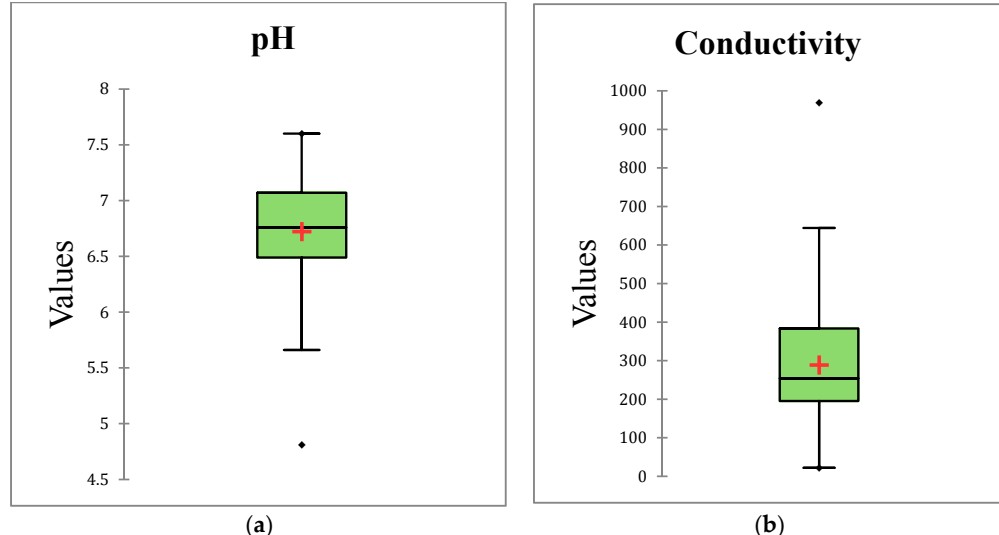

**Figure 2.** (**a**) pH variation. (**b**) Conductivity variation.

- **Turbidity**

Turbidity is one of the important physical parameters for water quality, defining the presence of suspended solids in water and causing the muddy or turbid appearance of a water body [9]. Very high values are found in the Massenya borehole (22.93 NTU). This recorded turbidity content exceeds the acceptable limit value for water intended for human consumption, which is 5 NTU [10]. It should be noted that the least cloudy waters are those of the cities of Ndjamena Koura, Klessoum, Aboussakine and Balabouda.

All values obtained comply with the Chadian national standard/WHO directive which stipulates that turbidity must be ≤ 5 NTU. In the towns of Massenya, Massenya Q La Paix, Massenya Djouboulio and Tourli I, the aquifers contain a clay or silty part. The turbidity of the water comes from the presence of suspended matter such as clay or silt, which gives the cloudy appearance to the water.

- **Total Dissolved Solids**

The level of dissolved solid or the level of dissolved salts varies significantly depending on the sampling point. The highest concentration of dissolved salts is observed in Ndjamena (379.0 mg/L).

The rate of dissolved salts varies according to the sampling points depending on the amount of dissolved matter in these waters.

- **Calcium ($Ca^{2+}$)**

Calcium is the major component of water hardness. In our water samples, calcium contents vary from 6 mg/L to 106.52 mg/L (Figure 3a). None of these waters therefore has a concentration greater than the WHO standard which is ≤200 mg/L.

This high content can be explained by the nature of the aquifer, which is partly made up of limestone. According to Potelon et al. [11], calcium is a metal extremely widespread in nature and in particular in limestone rocks in the form of carbonates ($CaCO_3$). It is encountered in almost all natural waters. The presence of $Ca^{2+}$ in the samples analyzed is directly linked to the geological nature of the terrain crossed. The values obtained comply with the Chadian/WHO water standards, which stipulate a maximum concentration of $Ca^{2+} \leq 200$ mg/L.

- **Sodium ($Na^+$)**

Sodium ion analysis of the water samples over all 83 sampling series gives levels ranging from 1.7 to 89 mg/L (Figure 3b), which is harmless to health, the limit value recommended by the WHO being ≤200 mg/L. Sodium is generally encountered in the form of sodium chloride salt.

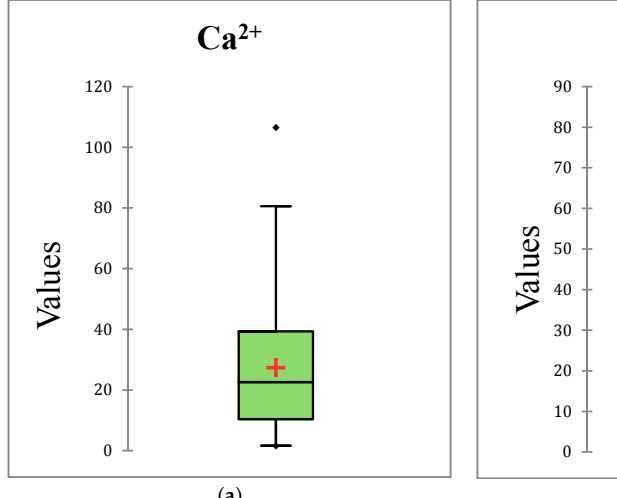
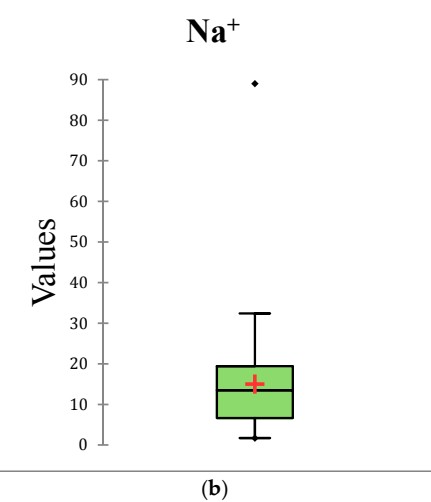

**Figure 3.** (**a**) $Ca^{2+}$ variation. (**b**) $Na^+$ variation.

The values obtained in this study comply with the Chadian National Standard/WHO Directive which stipulates that the content of sodium ions in drinking water must be ≤200 mg/L.

- **Potassium ($K^+$)**

The potassium values that we measured by flame atomic absorption spectrometry (Figure 4a) range from 0.1 to 7.88 mg/L. They are good below WHO standard.

The presence of these potassium concentrations is due to the rocks crossed. According to Potelon et al. [11], potassium is a natural element in waters where it's almost constant concentration usually does not exceed 10 to 15 mg/L. Furthermore, the values obtained in this study comply with the Chadian National Standard /WHO Directive which stipulates that the content of sodium ions in in drinking water must be ≤12 mg.

- **Chloride ($Cl^-$)**

Chlorides are important inorganic anions present in varying concentrations in natural waters and constitute an indicator of pollution. Their presence in groundwater may indicate anthropogenic contamination because of their existence in the urine as well as in the maintenance products.

According to Figure 4b, the contents vary between 0.65 and 67 mg/L. The maximum value, for its part, is encountered in the N'Djamena (Zozi) drilling.

The values obtained in this work comply with the Chadian National Standard/WHO Directive which indicates that the chloride ion content in drinking water must be ≤250 mg/L.

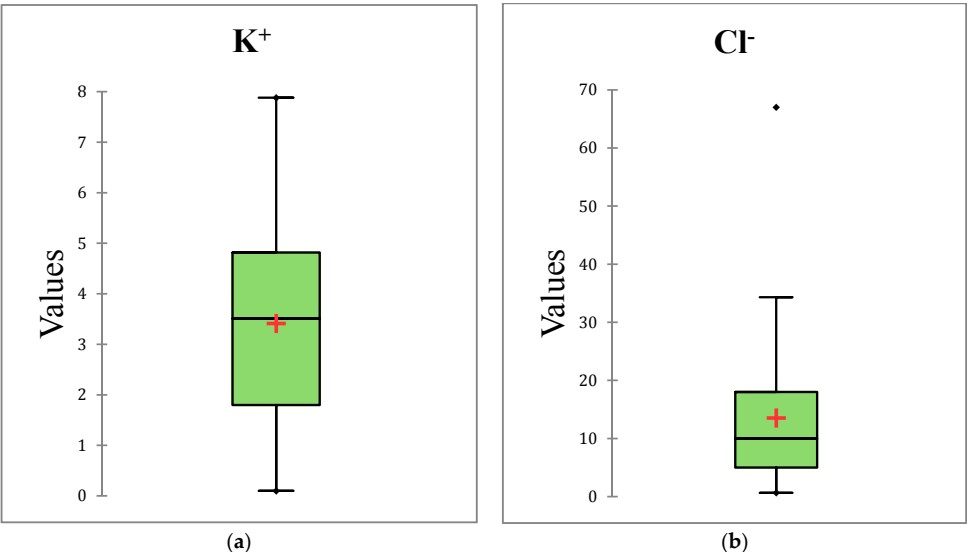

**Figure 4.** (**a**) $K^+$ variation. (**b**) $Cl^-$ variation.

- **Sulfate $SO_4^{2-}$**

Sulfate occurs naturally in water as a result of leaching from gypsum and other common minerals. Discharge of industrial wastes and domestic sewage tends to increase its concentration [12]. In this study area the values of this parameter in the waters studied are very variable as can be seen in Figure 5a. They oscillate between 0 and 58 mg/L, values below WHO standards (250 mg/L).

The presence of a high concentration of $SO_4^{2-}$ is due to the rocks crossed by the water. In this area the geology consists of a sedimentary rock, gypsum ($CaSO_4$). According to Meybeck et al. [13], the presence of sulfate in unpolluted waters invokes the presence of gypsum with a concentration varying between 0 and 58 mg/L. The waters analyzed comply with the National Standard/WHO Directive which provides for a concentration ≤250 mg/L.

- **Bicarbonate (HCO$_3^-$)**

Using the measured Complete Alkalimetric Title (TAC) values, we determined the bicarbonate content of the water samples (Figure 5b). There is no WHO standard for this element, but a high concentration of bicarbonates gives a salty flavor to water. The levels vary from 7.3–433.24 mg/L.

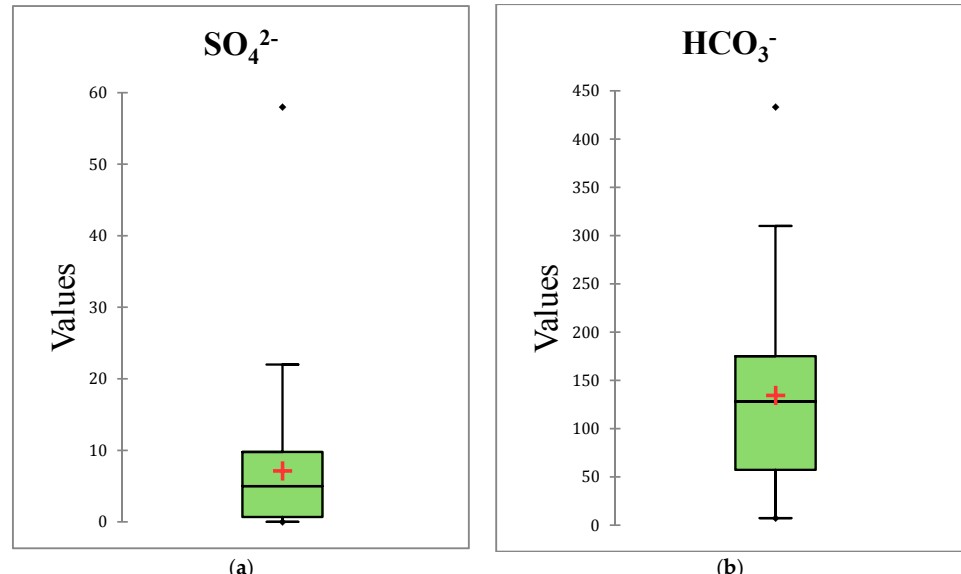

(a)　　　　　　　　　　　(b)

**Figure 5.** (**a**) Sulfate SO$_4^{2-}$ variation. (**b**) HCO$_3^-$ variation.

- **Magnesium (Mg$^{2+}$)**

Magnesium is the second most important element in water hardness after calcium. The concentration of magnesium varies according to the traversed terrain during infiltration [14]. Water rich in magnesium is beneficial for the consumer and has important intakes especially in cardiac and vascular function; it acts on cardiac excitability and vascular tone, contractility, reactivity and growth [15].

It is present in the waters of this region at levels ranging from 0.47–35.5 mg/L (Figure 6a). These levels are lower than the WHO accepted standard for magnesium, which is 50 mg/L.

- **Nitrate (NO$_3^-$)**

In the region studied, the nitrate content varied during the study period. It should be noted that, in most of the samples analyzed, there is not a high presence of nitrate ions (Figure 6b).

According to the Chadian national standard/WHO directive, the concentration of NO$_3^-$ must not exceed 50 mg/L. All the samples analyzed comply with the standard because the NO$_3^-$ concentrations obtained remain low and below the limits of said standard.

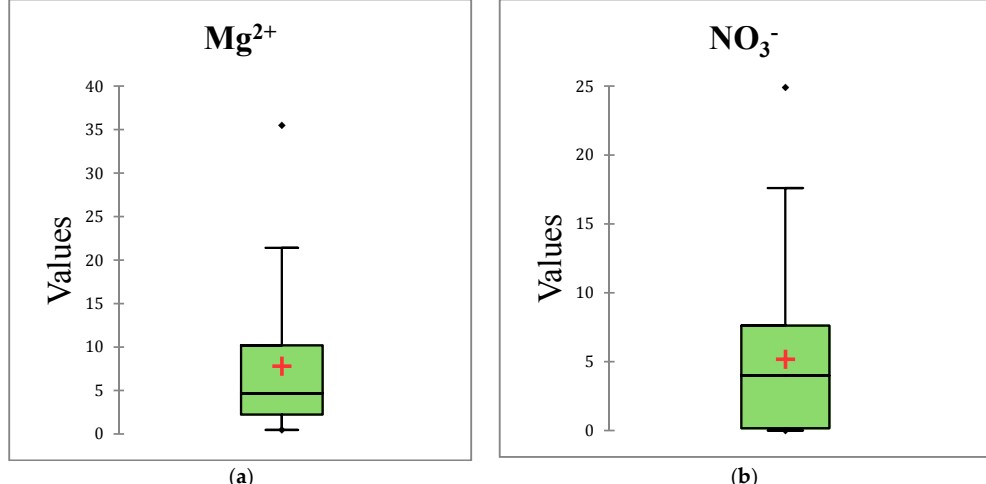

**Figure 6.** (**a**) $Mg^{2+}$ variation. (**b**) $NO_3^-$ variation.

- **Ammonium ($NH_4^+$)**

It should be noted that in most of the samples analyzed the presence of ammonium is remarkable (Figure 7a). According to the Chadian national standard/WHO directive, the $NH_4^+$ ammonium concentration must not exceed 1.5 mg/L. Some of the samples analyzed did not comply with this standard because the $NH_4^+$ ammonium concentrations obtained remain high and greater than the limits of said standard.

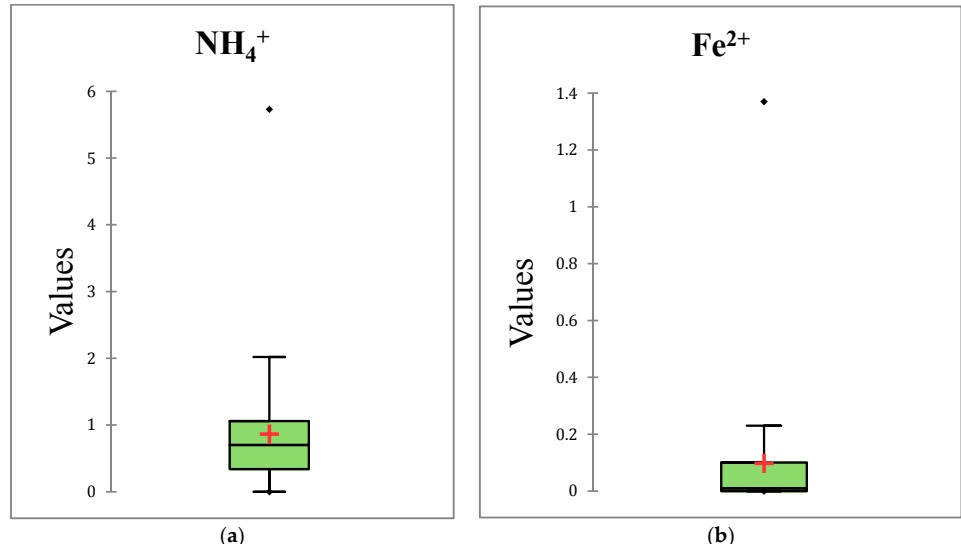

**Figure 7.** (**a**) $NH_4^+$ variation. (**b**) $Fe^{2+}$ variation.

- **Iron ($Fe^{2+}$)**

Iron is a fairly abundant element in rocks in the form of silicates, oxides and hydroxides, carbonates and sulfides. Iron is soluble in the $Fe^{++}$ ion state (ferrous ion) but insoluble in the $Fe^{+++}$ state (ferric ion). The value of the oxidation-reduction potential (Eh) of the medium therefore conditions its solubility and the iron content of the water. Captive aquifers isolated from exchanges with the surface see reduced conditions: their water is ferruginous. This dissolved iron precipitates in an oxidizing medium, in particular at the sources and at the outlet of the pipes.

The presence of iron in water can promote the proliferation of certain strains of bacteria which precipitate where the pipes corrode. The water is ferruginous in particular in certain layers found at

NDjamena and Ba-illi. A specific treatment is then necessary (precipitation in an oxidizing medium) [16]. The total iron contents in the study region vary from 0.00 mg/L–1.4 mg/L (Figure 7b), so the iron concentration in the region is above the recommended standard.

- **Hydro chemical facies**

The Piper diagram (Figure 8) shows the chemical facies of the set of water samples. It is composed of two triangles making it possible to modify the cationic facies and the anionic facies, and of a rhombus synthesizing the global facies.

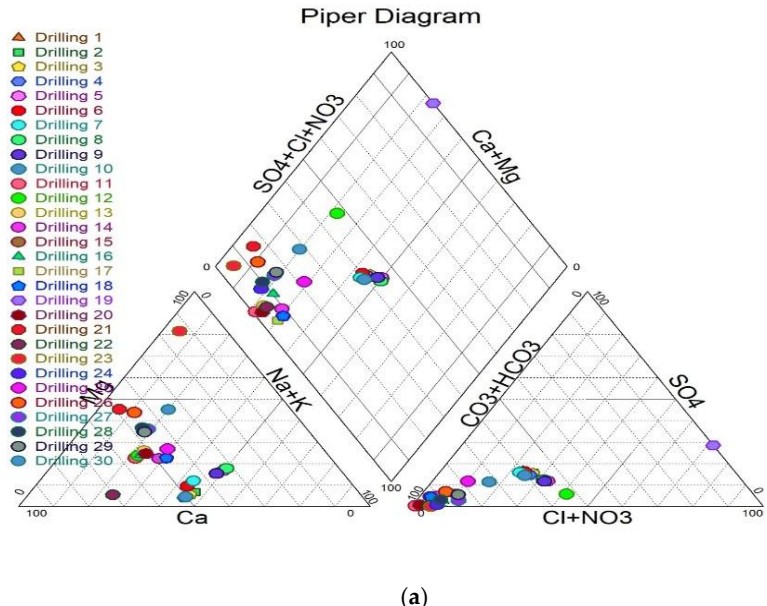

(**a**)

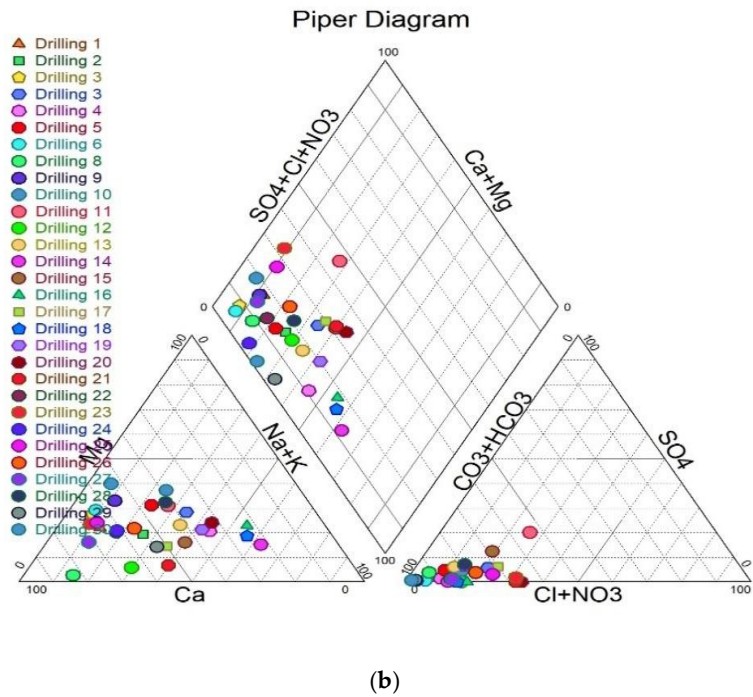

(**b**)

**Figure 8.** *Cont.*

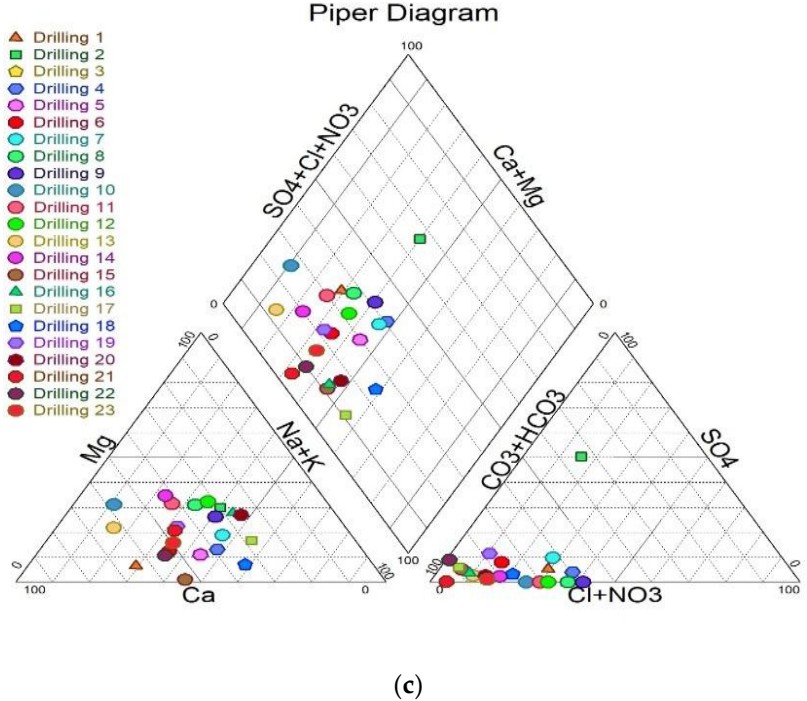

(**c**)

**Figure 8.** Paper Diagrams. (**a**) Paper diagram 14a of the samples analyzed shows 86.66% calcium and magnesian bicarbonate facies and 3.33% calcium and magnesian chloride and sulfate facies. (**b**) Paper diagram 14b of the samples analyzed shows 90% calcium and magnesium bicarbonate facies and 10% sodium and potassium bicarbonate. (**c**) Paper diagram 14c of the samples analyzed shows 82.60% calcium and magnesian bicarbonate facies, 8.69% sodium and potassium bicarbonate and 4.34% calcium and magnesian chloride sulfate.

Our Piper diagrams corresponding to the water samples from the eighty-three collection sessions are shown in Figure 8.

In Figure 8a–c we observe in the triangle of anions a predominance of bicarbonates, which reflects a facies bicarbonate. In the triangle of cations, there are no dominant ions, which translates as a mixed facies consisting of sodium, potassium, calcium and magnesian. These results are confirmed in the diamond where we see a global calcium and magnesian bicarbonate facies.

- **Principal Component Analysis**

The principal component analysis (PCA) method is widely used to interpret hydro chemical data. For data processing in principal component analysis, 12 variables were used, in this case pH, EC, $Ca^{2+}$, $Mg^{2+}$, $Na^+$, $K^+$, $Cl^-$, $SO_4^{2-}$, $NH_4^+$, $NO_3^-$, $HCO_3^-$ and $Fe^{2+}$

The total variance gives us an idea of the degree of information that each component represents.

The first component alone represents 34.19% of all the variable information, while it is these five variables that represent the total information.

If we regroup the five variables, we will have 74.18% of the information of all variables (Table 2), so there is no point in working on the whole set of variables because the variables (or components) are seen to be reliable.

**Table 2.** Extraction Method, Principal Component Analysis.

| | Total Variance Explained | | | | | |
|---|---|---|---|---|---|---|
| **Component** | **Initial Eigenvalues** | | | **Extraction Sums of Squared Loadings** | | |
| | **Total** | **% of Variance** | **Cumulative %** | **Total** | **% of Variance** | **Cumulative %** |
| 1 | 4.103 | 34.195 | 34.195 | 4.103 | 34.195 | 34.195 |
| 2 | 1.640 | 13.663 | 47.858 | 1.640 | 13.663 | 47.858 |
| 3 | 1.132 | 9.434 | 57.292 | | | |
| 4 | 1.069 | 8.905 | 66.197 | | | |
| 5 | 0.961 | 8.005 | 74.203 | | | |
| 6 | 0.867 | 7.225 | 81.427 | | | |
| 7 | 0.720 | 6.002 | 87.429 | | | |
| 8 | 0.455 | 3.795 | 91.224 | | | |
| 9 | 0.376 | 3.135 | 94.359 | | | |
| 10 | 0.353 | 2.942 | 97.301 | | | |
| 11 | 0.197 | 1.639 | 98.941 | | | |
| 12 | 0.127 | 1.059 | 100.000 | | | |

To obtain more information, there are two methods of interpreting the total variance. Either we choose variables which have a total greater than 1 and therefore in our case we have components 1,2,3,4 and 5 and/or we choose variables which, by making them cumulative, we can arrive at 80%. In our case the software has chosen five variables which have a total greater than or equal to 1.

- **Representation of the Variables on the Factorial Plane F1 and F2**

The correlation circle (Figure 9) shows that twelve variables taken into account in the PCA contribute to the definition of the factorial plane F1 × F2.

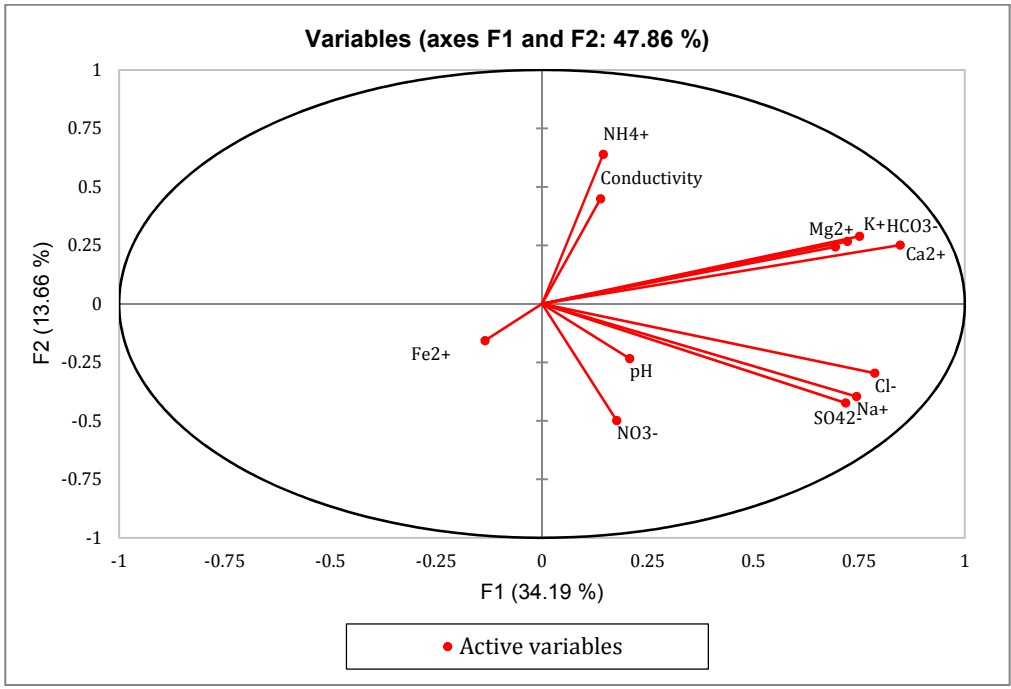

**Figure 9.** Correlation circle.

Analysis of the F1 and F2 factorial design shows that 47.86% variance is expressed. Factor 1 is expressed at 34.19% and represented mainly by: pH, EC, $Ca^{2+}$, $Mg^{2+}$, $Na^+$, $K^+$, $Cl^-$, $SO_4^{2-}$, $NH_4^+$, $NO_3^-$, but also by $HCO_3^-$ ions. The F2 axis is expressed at 13.66% and represented by iron.

Axis 1 is strongly positively correlated with electrical conductivity, bicarbonates, ammonium, potassium, magnesium and calcium. This axis expresses both the mineralization and organic pollution of the water. The conductivity measurement could therefore be sufficient to predict the quality of the water with regard to the above parameters.

This provides a simpler and faster way to monitor the water quality in a given area. The results obtained are similar to those of Ahmat [6] and Seid et al. [7] who showed that some boreholes in Ndjamena have concentrations much higher than the WHO standard.

Conversely, axis 2 is strongly negatively correlated with pH and nitrate and positively correlated with ammonium and electrical conductivity. This axis expresses less water mineralization compared to axis 1.

## 4. Discussion

The results corroborate with those of Alhabo, [6] and Seid et al. [7], reported in their research on the water quality of the city of NDjamena. The abnormal levels of iron and ammonium acquired in drilling water are similar to those we obtained during this research. However, the high iron and ammonium contents recorded respectively in certain localities constitute a major problem for the populations. Ammonium is the most reduced form of nitrogen and is the end product of the degradation of organic and inorganic matter in soil and aquatic environments [17]. The presence of ammonium in percolating groundwater indicates anthropogenic contamination. In addition, this element also comes from the bacterial activity of the soil, and from agricultural and industrial waste [17]. In fact, iron gives water an unpleasant metallic taste and a reddish color which could be linked to the deoxygenation of the water by organic activity in the soil and in the unsaturated zone [18].

The results of physicochemical analyzes of groundwater from some manual boreholes in the Chari Baguirmi region show that the water is slightly acidic with pH varying between 4.81 and 6.3.

In fact, in a humid tropical zone, this acidity would mainly come from the decomposition of plant organic matter, with the production of $CO_2$ in the first layers of the soil [19–21]. The predominance of hydrogen carbonate ions ($HCO_3^-$) associated with the $Ca^{2+}$ cation in the water sampled is thought to be a consequence of acid attack on the rocks and is also a characteristic of groundwater in the basement regions of Chad. These two elements are the origin of the calcium carbonate facies of the water samples.

The waters studied are weakly mineralized but with fairly high mineralization at Massenya (969 $\mu S.cm^{-1}$). This low water mineralization according to Youan Ta et al. [21], could be explained by the very poorly soluble nature of the host rocks.

## 5. Conclusions

Water resources are threatened by pollution which causes degradation of water quality. Surface pollutants can seep through the soil to the water table. The danger of pollution depends on the types and concentrations of the pollutants.

This study in the Chari Baguirmi region allowed us to highlight chemical pollution of the water and to assess the chemical content of its elements.

The physicochemical quality of the groundwater from 83 manual boreholes in the region of the Chari Baguirmi region was evaluated while performing analyzes of 12 physicochemical parameters.

This study showed that some boreholes are not recommended for consumption as drinking water. The parameters which downgrade this groundwater as non-consumable drinking water are iron and ammonium.

Among the samples analyzed, the water samples retrieved from Dourbali, Ba-illi, Bousso and Massenya are of good quality and can be used for human consumption. The other samples must be subjected to chemical treatments. To avoid the possibility of any health risk, it is recommended to treat water on a family scale by the use of hypochlorite, using a dropper, to extend the drinking water network in rural areas, and to design a sewerage network for wastewater disposal, garbage collection and protection of catchments.

Drilling water from this region of Chad is mainly characterized by a predominance of bicarbonate calcium and magnesian facies. The most contaminated wells are those located less than 5 to 15 m from potential sources of pollution such as direct toilets, full-bleed latrines, illegal dumps, poor well protection and the absence of a sanitation network.

**Author Contributions:** Funding acquisition, H.W.; Writing—original draft, A.B.B.; Writing—review and editing, X.J. All authors have read and agreed to the published version of the manuscript.

**Funding:** This study was financially supported by National R&D Plan Project and Study on the cooperation between the sustainable use of water resources and the integrated management of the lake basin in the Great Lakes basin in East Africa (No. 2018YFE0105900-2). We wish to thank the anonymous reviewers and editors for their thoughtful suggestion and careful work, which helped improve this paper substantially.

**Conflicts of Interest:** The authors declare no conflict of interest.

## Abbreviations

The following abbreviations are used in this manuscript.

| | |
|---|---|
| WHO | World Health Organization |
| PCA | Principal component analysis |
| NTU | Nephelometric Turbidity Unit |

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
