# Peer review of "Analysis and Control of the Physicochemical Quality of Groundwater in the Chari Baguirmi Region in Chad"

_water, doi:10.3390/w12102826_

Round 1
Reviewer 1 Report
Dear Authors
In part of chapter 1. Introduction [l 33] there was no background relating to the issues analyzed in the study. Authors should conduct an initial literature search. The introduction is modest and only applies to the region. L 61, 2.2. Methods of analysis - in this section, please provide more details on the analyzes used or connect to the point Laboratory analyzes [L 69]
In my opinion, the discussion [I 297] should contain a bit more references. I believe that the statistical analysis was carried out correctly at work. However, I missed other - simple tools, such as box charts, frames - whiskers. Their results are clear and easy to interpret.
Author Response
Dear reviewer!
We have taken into account the suggestions made for improving our article entitled "Analysis and control of the physico-chemical quality of groundwater in the Chari Baguirmi region in Chad" and modifications have been made in accordance with your suggestion.
Introduction: [49,50,51,52]
Result: [91,92,93,100,101,102,117,118,119,120] and all graphs have been redone in boxplots.
Conclusion: [329 342 343 344 345]
Best Regards
Reviewer 2 Report
The manuscript titled “Analysis and control of the physicochemical quality of groundwater in the Chari Baguirmi region in Chad” depicts an elaborate and comprehensive analysis of 12 physicochemical parameters in Chari Baguirmi region in Chad. The work is thorough and will provide important guidelines for future water quality management in that area. The writing and English grammar are also excellent and I enjoyed reading the manuscript. Therefore, I recommend acceptance in its current form or with minor corrections.
Couple minor corrections:
In many instances, there are paragraphs made of only one or two sentences. Please combine accordingly to make a paragraph with at least 4/5 sentences.
Figure 2 to Figure 12:
The plots quality should be improved. The x-axis data (samples) looks confusing and overlapping with one another. The gridlines should be removed or may be a few only. Y-axis label should also contain the units.
Author Response
Dear Reviewer!
Your suggestion has been taken into account and we have redone the graphics (2 to 12). We have also modified some paragraphs as you suggested.
Best Regards
Round 2
Reviewer 1 Report
Dear Authors,
Thank you for considering the suggestion.
Yours sincerely
Author Response
Thank you for your comments.